# Effect of *Lentinula edodes* on Morphological and Biochemical Blood Parameters of Horses

**DOI:** 10.3390/ani12091106

**Published:** 2022-04-25

**Authors:** Maria Soroko, Wanda Górniak, Paulina Zielińska, Aleksander Górniak, Karolina Śniegucka, Karolina Nawrot, Mariusz Korczyński

**Affiliations:** 1Institute of Animal Breeding, Wroclaw University of Environmental and Life Sciences, Chelmonskiego 38C, 51-630 Wroclaw, Poland; 112604@student.upwr.edu.pl; 2Department of Automotive Engineering, Mechanical Faculty, Wroclaw University of Science and Technology, Na Grobli 13, 50-421 Wroclaw, Poland; wanda.gorniak@pwr.edu.pl (W.G.); aleksander.gorniak@pwr.edu.pl (A.G.); 3Department of Surgery, Faculty of Veterinary Medicine, Wroclaw University of Environmental and Life Sciences, Plac Grunwaldzki 51, 50-366 Wroclaw, Poland; paulina.zielinska@upwr.edu.pl (P.Z.); sniegucka.karolina@gmail.com (K.Ś.); 4Department of Animal Nutrition and Feed Management, Wroclaw University of Environmental and Life Sciences, Chelmonskiego 38C, 51-630 Wroclaw, Poland; mariusz.korczynski@upwr.edu.pl

**Keywords:** horses, health, nutrition, thoroughbred, shiitake mushroom, supplement

## Abstract

**Simple Summary:**

Shiitake mushrooms (*Lentinula edodes*) have remarkable health properties that have been used in Far Eastern medicine for centuries. There is limited evidence that *L. edodes* has a direct effect on cell metabolism, with overall beneficial effects on animal health. As shiitake mushroom products appear to have a broad spectrum of effects, from changes in the immune response to changes in performance, a study was conducted to evaluate the effects of shiitake mushroom supplementation on the blood morphological and biochemical parameters in horses. This pilot study showed that there were single statistical differences between sessions between the supplemented and the control groups. The blood morphology showed that shiitake mushroom supplementation had an effect on white blood cells, hemoglobin, and hematocrit levels. In the biochemical analysis, shiitake mushrooms affected the levels of alkaline phosphatase, calcium, gamma-glutamyl transferase, bilirubin, and glucose and the albumin/globulin ratio. The observed differences between the supplemented and the control group sessions suggest that shiitake mushrooms may be a beneficial nutritional supplement for horses.

**Abstract:**

Shiitake mushrooms have been highly regarded as possessing enormous nutritive and medicinal values. No clinical studies have yet investigated the effect of shitake supplementation on the health of horses. The aim of this study was to evaluate the effect of shiitake mushroom supplementation on the morphological and biochemical blood properties in horses. A total of 17 adult horses were divided into two groups: supplemented and control. The supplemented group was fed 60 g of shiitake mushrooms per day for 5 months. Blood samples were collected in five sessions. Blood morphological analysis showed higher levels of lymphocytes in session 3 and monocytes in session 4 in the supplemented group. In addition, basophils, hemoglobin, and hematocrit were elevated compared to the control group. Biochemical analysis showed that the shiitake mushrooms affected a large number of parameters. In particular, alkaline phosphatase was found to be the most sensitive to shitake mushroom supplementation, for which the statistical differences were significant for sessions 2, 4, and 5. Furthermore, calcium was found to be affected by supplementation only in session 4, and gamma-glutamyl transferase in session 2. In addition, the bilirubin and glucose levels were lower in the supplemented group, and the albumin/globulin ratio was higher compared to the control group. The differences between the supplement and the control group in various sessions suggest that shiitake mushrooms are a beneficial nutritional supplement for horses.

## 1. Introduction

*Lentinula edodes*, commonly known as the shiitake mushroom, ranks second in the global mushroom market with regard to its nutritional value and its therapeutic application in the prevention or healing of multiple diseases and in balancing human diets [1].

Previous studies have proven the nutritional value of shiitake mushrooms. This is due to their high content of biologically active components, including proteins, carbohydrates, and, in particular, bio-active polysaccharide complexes such as β-D-glucan, heteroglucan, xylomannan, lentinan, eritadenine heteroglucan, and lentinan. Free sugars such as arabinose, arabitol, mannose, mannitol, trehalose, and glycerol are also present [2,3]. Shiitake are also rich in vitamins B2, B12, D2 [2], and micro- and macro-elements (K, P, Mg, Ca, Zn, Cu, Mn, Se, and Fe) [4,5,6]. They are low in cholesterol and total fat but contain a high proportion of unsaturated fatty acids [7].

Recent studies on humans have shown the medicinal attributes of shiitake in reducing LDL cholesterol level, improving liver function, lowering arterial blood pressure, and influencing the immune system [8,9]. Furthermore, shiitake mushrooms have been used to treat tumor cells and have a big potential for fighting against various human diseases [10,11].

The effect of shiitake mushrooms on animal health has not been fully investigated. Recent studies on rats indicated an effect from shiitake supplementation on the growth of specific gut microbes. The gut microbiome of the supplemented group had a higher species richness, characterized by the increased abundance of *Clostridium* and *Bacteroides* spp. compared to the control groups. It was found that manipulation of the gut microbiota through the administration of *L. edodes* could manage dyslipidemia [12]. Research based on mice indicated that mushroom supplementation significantly lowered serum total cholesterol, LDL cholesterol, and triglycerides, which could help in the regulation of lipid metabolism [13]. Studies on poultry indicated higher bodyweight gain and increased cecal microbial counts in the supplemented group, which showed shiitake’s potential as a prebiotic that leads to shifts in the intestinal microbial populations of supplemented chickens [14].

Although the studies investigating the effects of *L. edodes* are limited, there is evidence to suggest that supplementation with shiitake mushrooms has a direct effect on cellular metabolism, with overall beneficial effects on animal health. However, no data are available regarding the biological effects of supplementation with these fungi specifically on horses. Therefore, the objective of our study was to evaluate the effect of shiitake mushroom supplementation on the morphological and biochemical blood properties in horses. It was hypothesized that supplementation with shiitake mushrooms would positively influence and improve the morphological and biochemical blood profiles in horses.

## 2. Materials and Methods

Ethical approval was obtained from the Local Ethical Committee for Animal Experiments in Wroclaw prior to data collection in June 2019 (protocol code 036/2019/P1).

### 2.1. Animals

The research was conducted on a group of 17 riding-school, thoroughbred adult horses (13 geldings and 4 mares), 4–10 years old, with a BCS of 5.0–5.5 [15] and an average body weight of 492 ± 58 kg. The horses were clinically healthy and in good condition. A clinical history was obtained from the case files to determine whether the horses had had any health issues in the previous three months. A standard physical examination was performed by an experienced equine clinician to confirm any health problems. The examination included rectal temperature, oral mucous membrane color and capillary refill time, mandibular lymph node assessment, heart auscultation and rate, respiratory tract auscultation and respiratory rate, intestinal borborygmi, and digital pulsation. The horses had clinical examination every 4 weeks during the period of the study. The animals were housed in the same stable, in individual box stalls bedded on straw, at the riding club in Wroclaw (Poland). The horses that qualified for the study were kept in box stalls of dimensions of 3 × 3 m overnight and on pasture for half a day (from 6.30 a.m. to 1.00 p.m.). After returning from the pasture, the horses had access to hay and water ad libitum. The animals were fed concentrated feed three times a day at 6.00 a.m., 1.00 p.m., and 6.30 p.m.

The horses consumed a constant basal diet, which was based on the nutrient requirements of horses, throughout the study [16]. The diet consisted of 3 kg of meadow hay three times a day, a mixture of beet pulp concentrate (AgroVital BASIC, Agro-Vital, Niewodnica Koscielna, Poland—ingredients composition: unmolassed beet pulp, granulated alfalfa dry, dried apple, evening primrose and linseed oilcake, buckwheat husk, soybean oil, sodium chloride, calcium phosphate, magnesium oxide) and compound feed based on raw materials rich in structural carbohydrates (AgroVital CONTROL, Agro-Vital, Niewodnica Koscielna, Poland—ingredients composition: dried alfalfa, rice bran, barley, sunflower oilcake, linseed oilcake, dried apple husk, buckwheat husk, black cumin and evening primrose oilcake, beet pulp, sunflower oil, magnesium oxide, calcium phosphate, sodium chloride, and seaweed calcium). An amount composed of 500 g of beet pulp concentrate and 500 g of compound feed based on raw materials rich in structural carbohydrates was divided into three feedings.

Horses were used for pleasure riding 5 times a week for 1–2 h a day. During the research period all the horses continued to participate in riding classes.

### 2.2. Experimental Procedure

The shiitake mushrooms were cultivated by the Ecological Mushrooms Farm in Kalisz, Poland. Selected mushrooms were air-dried at about 40 °C for approximately 3 days and ground into powder. The mushroom powder was then granulated with additives. The 100 g of granulate contained: 30 g of dried shiitake mushrooms, 50 g of barley, 10 g of wheat bran; 4 g of corn, 4 g of lucerne, and 2 g of molasses.

The selected research horses were randomly divided into 2 groups: the supplemented group (G1) included 9 horses (4 aged 10 years, 3 aged 9 years, 2 aged 4 years; 7 geldings and 2 mares) and the control group (G0), which included 8 horses (2 aged 10 years, 2 aged 9 years, 1 aged 8 years, 2 aged 7 years, and 1 aged 4 years; 6 geldings and 2 mares). The supplemented group was fed with additional 200 g of granulate, containing 60 g of shiitake mushroom every day for 112 days (from August until December 2020), once a day at the afternoon feeding. The supplementation started on the first day of the experiment (day 1). The amount of shiitake supplementation was based on the producer’s recommendations. The control group was given supplementation granules without the addition of shiitake mushrooms throughout the study period.

### 2.3. Blood Collection and Analysis

Blood collection from each horse was performed in five sessions (sessions 1–5/S1–S5) at intervals of 4 weeks. The blood collection was on day 1 (session 1), day 28 (session 2), day 56 (session 3), day 84 (session 4), and day 112 (session 5). In each session, the blood was taken at rest, before the morning feeding, by puncture of the external jugular vein (vena jugularis externa) using a sterile BD Vacutainer^®^ system into K2-EDTA tubes for hematological analyses, plain tubes for serum biochemistry, and fluoride tubes (sodium fluoride 15 mg/mL/EDTA 3.0 mg/mL) to assess lactic acid concentration (lactic acid, mmol/L) (Plymouth, UK). All blood samples were examined within a maximum period of 1.5 h after collection. For venipuncture, 20 G 1/2” needles were used.

Blood morphological analyses were performed using a Sysmex XN-1000 hematology analyzer (Sysmex America, Inc., Lincolnshire, IL, USA). The following parameters were assessed: white blood cells (WBC, 10^9^/L), neutrophils (NEU, 10^9^/L, %), lymphocytes (LYM, 10^9^/L, %), monocytes (MONO, 10^9^/L, %), eosinophils (EOS, 10^9^/L, % ), basophiles (BASO, 10^9^/L, %), red blood cells (RBC, 10^12^/L), hemoglobin (HGB, mmol/L), hematocrit (HCT, L/L, %), mean corpuscular volume (MCV, fL), mean corpuscular hemoglobin (MCH, fmol), MCHC (mean corpuscular hemoglobin concentration, mmol/L), and platelets (PLT, 10^12^/L).

The plain tubes were centrifuged (2800 rpm. for 5 min) using a benchtop Rotanta 460 centrifuge (Andreas Hettich GmbH & Co. KG, Tuttlingen, Germany), and the serum was aspirated. The following biochemical parameters were estimated: albumin (g/L), alkaline phosphatase (AP, U/L), aspartate aminotransferase (AST, U/L), total protein (g/L), total bilirubin (μmol/L), chlorides (mmol/L), cholesterol (mmol/L), creatine kinase (CK, U/L), phosphorus (P, mmol/L), glutamate dehydrogenase (GLDH, U/L), glucose (GLUC, mmol/L), gamma-glutamyl transferase (GGTP, U/L), creatinine (μmol/L), lactate dehydrogenase (LDH, U/L), magnesium (Mg, mmol/L), urea (mmol/L), potassium (K, mmol/L), sodium (Na, mmol/L), triglycerides (TGL, mmol/L), calcium (Ca, mmol/L), globulins (g/L), and the albumin/globulin ratio (mmol/L). All the parameters were determined by an AU680 clinical chemistry analyzer (Beckman Coulter Inc., Brea, CA, USA). For all measurements, Sysmex (Sysmex America, Inc., Lincolnshire, IL, USA) and Backman (Beckman Coulter Inc., Brea, CA, USA) reagents, calibrators, and standards were used. Only the GLDH concentrations were determined by Randox diagnostic veterinary reagents (Randox, Crumlin, UK).

### 2.4. Data Analysis

The entire investigation was performed with a significance level of α = 0.05. The Shapiro–Wilk and Kolmogorov–Smirnov normality verification proved that the measurement did not originate from the Gaussian distribution, and therefore, non-parametric tests were used. The difference between the research group (G1) and the control group (G0) was verified by means of the Mann–Whitney U significance test. Each session was considered separately. Only the differences between the two groups were compared, and therefore, the groups were considered as independent. The Friedman ANOVA test was used to test the significance of differences of the blood parameters in all the sessions in the research group. Subsequently, for significant parameters a post hoc (Conover–Iman test) was used to verify the differences between the sessions. The Statistica software (v. 13.3, StatSoft Inc., Tulsa, OK, USA) was used for all calculations.

## 3. Results

### 3.1. Morphological Analysis of Blood

The results of the comparison between the supplemented and the control groups for morphological analysis are presented in Table 1. A significant difference was observed for LYM in session 3, as well as for MONO and MONO% in session 4. Closer inspection of the *p*-value suggested that during session 3 the differences between the groups were the highest. Only LYM was statistically important, but there are other cases whose *p*-value was close to the boundary of 0.05, such as, for example, LYM% (*p* = 0.05415), EOS% (*p* = 0.054), and BASO and BASO% (*p* = 0.07042, *p* = 0.05824, respectively).

In order to verify the influence of the mushroom supplementation throughout the study, a Friedman’s ANOVA was used. The same horses were compared in a different session; hence, the variables were considered as dependent. Moreover, taking into consideration the main goal of the investigation, only the supplemented groups are presented in all the Friedman’s ANOVA tables.

Only 10 measurements of the entire morphological set differed significantly throughout the study. The parameters which were statistically different through the sessions and the corresponding *p*-values are shown in Table 2. Considering the medians, it was apparent that there was no increasing tendency (Table 2). One exception was HCT, which increased with respect to time. Furthermore, statistically significant differences between the sessions also differed for each parameter. The greatest difference between the sessions was found for LYM%. Here, session 5 (S5) was different from all other sessions. Similarly, session 4 (S4) differed from the other sessions, except for session 1 (S1). Session 3 (S3) differed in this case from all the other sessions, except for session 2 (S2).

In addition, there was a continuous increase in the mean HGB value in the supplemented group throughout the duration of the study (S1—111 10^12^/L; S2—114 10^12^/L; S3—116 10^12^/L; S4—119 10^12^/L; S5—122 10^12^/L) and a decrease in the values of the parameters related to white blood cells (WBC, NEU, MONO, EOS) (Appendix A, Appendix A). In the control horses, the morphological blood analysis trends were similar but significantly lower than in the group of supplemented horses. Importantly, in G1, the level of the BASO value throughout the research period ranged from 0.06–0.08 10^9^/L, and in the G0 from S2, it was at the level of 0.04–0.05 10^9^/L.

### 3.2. Biochemical Analysis of Blood

The analysis of the differences in the biochemical factors is presented in Table 3. The AP level represented the highest differences between the control and the supplemented groups from session 2. Session 3 did not constitute a statistically significant difference for AP; however, the *p*-value was close to the boundary of 0.05. The most significant difference was in session 5.

The significant biochemical parameters are shown in Table 4. In this case, most of the parameters from the entire set were significant for the adopted α of 0.05. Moreover, session 3 differed from the others with respect to the fact that most of the parameters and the median increased until session 3 and fell afterwards. Consequently, the post hoc analysis detected the highest number of differences between the sessions. Additionally, in the case of the biochemical analysis, the most significant parameters (albumin/globulin ratio; *p* = 0.000042) had the highest number of differences between the sessions.

Furthermore, the group of horses supplemented with shiitake mushrooms showed a decrease in the mean values of cholesterol, glucose, and bilirubin compared to the control group (Appendix A).

## 4. Discussion

Numerous studies have shown that supplementation with shiitake mushrooms has various benefits, including immunomodulatory, hypocholesterolemic, and anti-tumor effects on both animals and humans [17,18]. However, no research has been conducted to date on the effects of mushroom supplementation on healthy horses. Our pilot study is the first to indicate the influence of shiitake mushrooms on the blood parameters in horses, which were within the correct reference range dedicated to equines. There were single statistical differences among sessions between the supplemented and the control groups.

For the morphological parameters, the group of supplemented horses had statistically higher levels of LYM in session 3 and MONO in session 4 compared to the control group. In addition, the supplemented group had BASO values higher throughout the research period compared to the control group. Elevated white blood cells are essential in mediating immune and inflammatory responses. Many studies have indicated the immune-enhancing effects of mushrooms in other species [19,20]. In a study based on elks, the group supplemented with spent mushroom substrate had significantly higher levels of blood monocytes compared to control animals [21]. Dalloul et al. [22] investigated the immune-potentiating effect of a mushroom lectin on poultry cell-mediated immunity and protection against coccidiosis. Their results from utilizing the mushroom lectin included effective growth promotion and immune stimulation in poultry with coccidiosis. A study based on broilers detected no changes in heterophil and lymphocyte percentages in the group supplemented with shiitake mushrooms [20]. For many horses, the demands of intensive training, competition, and traveling and the accompanying stress put their immune systems under the pressure. Keeping a horse’s immune system strong is crucial to maintaining health and preventing infection and stress-induced immunodeficiency.

The HCT and HGB levels in our group of supplemented horses increased throughout the study. In the literature, mushrooms have been indicated as a source of iron, which is essential for the synthesis of HGB. These results appeared to be consistent with the study presented by Park et al. [21], where the level of HGB and HCT in elks supplemented with spent mushroom substrate was significantly increased. Another recently conducted study indicated that feeding products with 10% and 20% dried shiitake to rats with iron deficiency resulted in an increase in blood HGB concentration [23].

Our current study indicated a tendency towards cholesterol decrease in the research group. This result is in agreement with the study presented by Xu et al. [24] where the administration of polysaccharides from shiitake mushrooms significantly reduced serum total cholesterol in rats. In previous studies based on rats, it was found that the addition of mushrooms into the diet of experimental animals effectively prevented the development of hypercholesterolemia and the accumulation of cholesterol in the liver [25]. Various studies have confirmed that the investigated mushrooms can lower blood pressure and free cholesterol in plasma [2,26].

Bilirubin and AP were lower in the supplemented group compared to the control group. Both parameters evaluate the functional ability of the liver, and the increased activity of bilirubin and AP may indicate liver damage or be associated with bone injuries [27]. Increased bilirubin levels can be caused by feed deprivation [28], and AP elevation is associated with abscess formation [29] and has also been described in horses with young age [30]. In the study presented by Park et al. [21], concentrations of AP and bilirubin were found to be similar between a group of elks supplemented with spent mushroom substrate and the controls.

In addition, differences in glucose levels were found between our groups. Glucose supplies energy from food to all the cells in the body and is an indicator of the energy status of the horse. In the conducted study, the horses supplemented with *L. edodes* had lower glucose concentrations. Shiitake mushrooms contain biologically active polysaccharides, which belong to the group of β-glucans, which can prevent high blood glucose and cholesterol levels or insulin resistance [31,32]. Similar results were obtained in the study of an antidiabetic drug and the antioxidant activity of β-glucan from shiitake mushrooms in mice [33]. The phenomenon of lowering blood glucose levels due to the administration of shiitake mushrooms may be of importance for horses suffering from hyperglycemia.

The TGL level was significantly lower in our supplemented group compared to the control group only in the first session. Previous studies based on rats found that adding shiitake mushrooms to a high-energy diet can significantly lower plasma TGL [34].

The present study also indicated the influence of mushrooms on the level of macro-elements, including Na and Ca. The supplemented group had lower Na and higher Ca compared to the control group. None of the previous studies indicated the influence of mushrooms on the macro-element level. However, research on the assessment of the chemical composition of dried shiitake estimated low contents of Na [35] and Ca [4]. The study presented by Park et al. [21], found that the inclusion of spent mushroom substrate into the diet of elks did not affect the Na or Ca concentrations in the blood.

There is currently a lack of research regarding shiitake supplementation in horses, and previously reported results in other species cannot be directly compared. Nevertheless, the findings of the presented investigation support the results of previous research based on mushroom substrate supplementation conducted on different species [21,36,37]. Further trials may benefit from this study design, focusing on the amount of the shitake supplementation and selecting for study blood parameters which were the most influenced by supplementation. Future studies should focus on detecting shitake supplementation on the horses of the same age and fitness. The major limitation of our study was variety in the research group of horses, which presented different age and fitness levels, which could affect shitake supplementation and introduce variability into the blood parameters. It also has to be considered that our study was a small project involving only 17 horses, and this may have limited the statistical power of the study. We adopted a study design which averaged the measurements taken in five sessions to reduce the variability in the data. Small differences in the morphological and biochemical results between groups could be associated with changing the feeding habits during the research period. The horses had access to the grass on the pasture for the first three months of the study, which could have had a significant influence on the blood parameters (particularly in session 3).

## 5. Conclusions

The presented pilot study showed that there were single statistical differences between the sessions in the supplement and the control groups. With growing evidence that shiitake mushrooms improve human and animal health, more research is needed to indicate the beneficial effects of *L. edoes* on horse health.

## Figures and Tables

**Table 1 animals-12-01106-t001:** *p*-Value representing differences between supplemented (G1) and control group (G0) for morphological analysis.

Parameter, Unit	Session 1 (S1) G0 vs. G1	Session 2 (S2) G0 vs. G1	Session 3 (S3) G0 vs. G1	Session 4 (S4) G0 vs. G1	Session 5 (S5) G0 vs. G1
WBC, 10^9^/L	0.53167	0.96163	0.96163	0.88523	0.96163
NEU, 10^9^/L	0.80989	0.31232	0.80989	0.73628	0.66501
NEU%	0.9616	0.22905	0.17767	0.80989	0.41312
LYM, 10^9^/L	0.22905	0.16268	0.03856	0.41341	0.33563
LYM%	0.56346	0.11235	0.05415	0.36065	0.22905
MONO, 10^9^/L	0.28806	0.92268	0.53116	0.02978	0.56298
MONO%	0.06667	0.73597	0.21012	0.03008	0.66306
EOS, 10^9^/L	0.9233	0.73534	0.08251	0.96156	0.49873
EOS%	0.44114	0.88495	0.054	0.96156	0.47021
BASO, 10^9^/L	0.34855	0.30541	0.07042	0.73439	0.35286
BASO%	0.37877	0.20532	0.05824	0.80874	0.35679
RBC, 10^12^/L	0.50006	0.53167	0.36065	0.77256	0.96163
HGB, 10^12^/L	0.73597	0.66462	0.77269	0.22762	0.31024
HCT, L/L	0.80944	0.80967	0.73612	0.24792	0.36035
MCV, fL	0.63001	0.22905	0.22905	0.41312	0.31084
MCH, fmol	0.35976	0.50006	0.49873	0.28925	0.28954
MCHC, mmol/L	0.80967	0.12297	0.66443	0.84693	0.24471
PLT, 10^12^/L	0.41255	0.88523	0.70014	0.77256	0.19201

**Table 2 animals-12-01106-t002:** Results of Friedman’s ANOVA and post hoc test for morphological analysis for the supplemented group.

Parameter, Unit	*p* Value	Median				
		Session 1 (S1)	Session 2 (S2)	Session 3 (S3)	Session 4 (S4)	Session 5 (S5)
LYM%	0.00032	26.3	27.5	25.1	31.7	32.2
LYM, 10^9^/L	0.001416	2.09	2.09	1.94	2.09	2.39
MCV, fL	0.003918	46.4	47.3	46.7	47.6	47.4
NEU%	0.004138	61.2	62.8	64.5	60.3	58
MONO, 10^9^/L	0.013515	0.48	0.4	0.47	0.38	0.41
MCHC, mmol/L	0.01415	354	346	354	345	351
HCT, L/L	0.01856	0.323	0.332	0.327	0.343	0.352
BASO, 10^9^/L	0.02678	0.08	0.07	0.06	0.07	0.06
NEU, 10^9^/L	0.034203	4.95	4.36	4.85	4.26	4.42
WBC, 10^9^/L	0.035501	8.45	7.94	7.76	6.78	7.43

**Table 3 animals-12-01106-t003:** *p*-Value representing differences between supplemented (G1) and control group (G0) for biochemical analysis.

Parameter, Unit	Session 1 (S1) G0 vs. G1	Session 2 (S2) G0 vs. G1	Session 3 (S3) G0 vs. G1	Session 4 (S4) G0 vs. G1	Session 5 (S5) G0 vs. G1
Albumin, g/L	0.268179	0.413408	0.847112	0.247924	0.134874
AP, U/L	0.312322	0.023742	0.082321	0.038562	0.014079
AST, U/L	0.665006	0.736277	0.961627	0.736277	0.360645
Total protein, g/L	0.101667	0.288951	0.312322	0.066669	0.135834
Bilirubin, µmol/L	0.596416	1.000000	0.531416	0.923295	0.698371
Chlorides, mmol/L	0.162164	0.595964	0.594829	0.074336	0.311133
Cholesterol, mmol/L	0.596416	0.091995	0.247343	0.809894	0.268179
CK, U/L	0.413121	0.311728	0.885234	0.847297	0.162937
P, mmol/L	0.531164	0.736121	0.228478	0.809894	0.563225
GLDH, U/L	0.736277	0.340763	0.268473	0.961627	0.386186
Glucose, mmol/L	0.193932	0.359465	0.961627	0.312026	0.596416
GGTP, U/L	0.247924	0.048403	0.700137	0.162937	0.083077
Creatinine, µmol/L	0.177667	0.700137	0.500319	0.360645	0.772695
LDH, U/L	0.531668	0.885234	0.268473	0.360645	0.736277
Mg, mmol/L	0.411394	0.735496	0.209280	0.059657	0.629587
Urea, mmol/L	0.091995	1.000000	0.961627	0.961603	0.193932
K, mmol/L	0.961627	0.531668	0.135595	0.359169	0.335628
Na, mmol/L	0.440858	0.066837	0.385602	0.885094	0.562021
TGL, mmol/L	0.059815	0.663062	0.660883	0.562021	0.177134
Ca, mmol/L	0.192014	0.497402	0.309939	0.014020	0.469942
Globulins, g/L	0.413121	0.531164	0.360351	0.135834	0.385894
Albumin/globulin ratio, mmol/L	0.772695	0.736277	0.440577	0.531416	0.735965
Lactic acid, mmol/L	0.563225	0.468573	1.000000	0.247924	0.440577

**Table 4 animals-12-01106-t004:** Results of Friedman’s ANOVA and post hoc test for biochemical analysis for the supplemented group.

Parameter, Unit	*p* Value	Median				
		Session 1 (S1)	Session 2 (S2)	Session 3 (S3)	Session 4 (S4)	Session 5 (S5)
Albumin/globulin ratio, mmol/L	0.000042	1.0	1.0	1.0	1.1	1.1
Glucose, mmol/L	0.000142	5.2	4.7	4.6	4.8	4.8
Cholesterol, mmol/L	0.000249	2.0	1.8	1.8	1.9	1.9
Bilirubin, µmol/L	0.001056	20.1	20.3	24.4	19.2	16.1
Total protein, g/L	0.00115	67.2	67.0	69.8	66.6	62.2
Globulins, g/L	0.00178	34.0	33.4	36.3	34.4	31.0
CK, U/L	0.001803	256.0	376.0	371.0	322.0	331.0
Mg, mmol/L	0.002783	0.9	0.8	0.8	0.9	0.8
LDH, U/L	0.005662	354.2	361.0	436.4	342.1	318.9
Albumin, g/L	0.007014	31.7	32.8	34.8	33.7	33.0
Urea, mmol/L	0.00785	6.6	5.7	7.0	6.1	6.2
P, mmol/L	0.008039	0.9	1.2	1.0	1.0	0.9
Lactic acid, mmol/L	0.010926	0.7	1.0	1.2	0.8	0.8
GGTP, U/L	0.03498	12.6	12.4	13.2	13.4	14.6
Ca, mmol/L	0.036314	3.1	3.1	3.1	3.2	3.1
AP, U/L	0.049519	209.0	237.0	199.0	195.0	210.0

## Data Availability

The data presented in this study are available on request from the corresponding author. The data are not publicly available for privacy reasons.

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
