# Peer review of "Effect of Lentinula edodes on Morphological and Biochemical Blood Parameters of Horses"

_animals, 2022, doi:10.3390/ani12091106_

Round 1

Reviewer 1 Report

General comments and suggestions for Authors:

You present a very interesting study which opens an important research area of veterinary medicine and pharmacology, deserving further development!

Abstract:

P1/line 32:  phrase “horses’ healthy” – do you mean “horses’ health” or “healthy horses”? Please, use a proper phrase!

P1/line 39: phrase “a greater number” should be changed to “a large number”

 P1/line 44-45: the differences were found in different sessions, so you should complete the sentence (like: …. and control group in various session suggest that …)

Introduction:

P2/line 56: delete comma (before [2,3])

P2/line 68: “in regulating” should be changed to “in regulation of”

Materials and Methods

P2, lines 85-90: The description of selection criteria is not clearly written! As far as I understand, you reviewed clinical history of horses to exclude animals with any clinical history in last 3 months before experiment. Thereafter, a standard clinical examination was made in selected horses. So, were any horses excluded / not selected due to health problems? Or did you select a group of horses and thereafter controlled their health status?

P3/line 113: spelling error (The shiitake …)

P3/lines 11-120:  I suggest you to rewrite the sentence (for example: The group of selected research horses were randomly divided into 2 groups: the first supplemented group (G1) included 9 horses and the second was a control group (G0), which included 8 horses.)

P3/line 120: “supplementation” change to “supplemented” (to unify expression with the other text); delete “an” before “additional”;

P3/lines 126-128: The sampling protocol is not clear. As far as I understand, you treated horses for 5 months (cca 150 days) and sampled them 5 times (4 weeks apart x 28 days = 128 days). The numbers of days don’t fit together considering your description!  When was the first sampling? At the day of the treatment onset or some days later? When was the last sampling – at the end of feeding? Please, explain sampling flow more transparent!

P3/lines 128-129: I suggest you to change the sentence as follows: “In each session blood collection was taken at rest, before the morning feeding . Blood samples were taken by puncture of the external jugular vein (vena jugularis 129 externa) using …”

P3/ lines 135-150: I suggest you to use SI units for measured parameters, for instance:

  • For litres “L” instead of “l” in majority of cases;
  • For WBC and leucocyte subsets 106/L (instead of G/l or G/L),
  • For RBC and platelets 1012/L (instead of T/L).

these units are not commonly accepted in haematology! (see https://onlinelibrary.wiley.com/doi/full/10.1111/ijlh.12563)

P3/line 140: ad “mean”, as follows: MCHC (mean corpuscular …)

Results:

General comment: Although transparency of Tables 1 to 4 was improved compared to original (first) version of MS, I regret removal of tables with median values of measured parameters (Tables 3 and 6 of the original/first MS version)! These values would clearly demonstrate changes of absolute values measured in treated and control horses throughout the experiment, actually representing basis for statistical analyses! I suggest you to publish the tables with measured values online alongside the manuscript as supplementary materials!

Tables 1 to 4: Please, insert titles of the Column 1 (Parameter, unit)

P3/lines 196-202 and Tables 1 - 4: see comments for P3/ lines 135-150!

 Discussion:

General comment:

  • Were the measured values within normal ranges for horses? Did you find any deviations from the normal? Please, find a proper location in the discussion to indicate this!
  • You cite several studies which indicated immune-enhancing or other effects of mushrooms. Could you suggest practical use of this effect in horses (like you described for Fe and hyperglycemia) in prevention or treatment of certain pathological states in horses?
  • Can you expect different reactions of (at least some) measured parameters in sick horses (considering several pathological processes are reflected in changes of various haematological and blood biochemistry parameters)?

P7/Line 231: I believe you should emphasize that the investigated horses were healthy!

P7/line 295: Can you expect different responses in sick horses than in healthy ones?

Reviewer 2 Report

The article about “ Effect of Lentinula edodes on morphological and biochemical blood parameters of horses” is interesting for the professional sport training. Especially for trainers, owners and veterinary practitioners but also for the enthusiasts. The present paper is interesting. Also it is important in connection to translational medicine.

However, it needs several correction.

First of all the Authors should rewrite several paragraphs because sometime it is hard to follow.

Introduction

It should be more precise because there are several repetitions without specifics. For ex. please add information how glucans influences on immune funcitions. What is a mechanism of xylomannan or lentinan action?

L63 – what kind of infection and metabolic diseases?

Authors have to be more specific. How strong the changes were expressed? In my opinion information that sth has changed is not enough.

Materials and Methods

Once again authors should be more specific. Did horses became from one stable? Also the mean age and gender for animals in each group should be mentioned, especially when horses had 4-10 years.

Did all parameters measured at equipment dedicated for equine species?

Results

The results presentation is not reader friendly. It is hard to follow.

Discussion

I am not sure that the LYM, MONO, basophils, eosinophils, Ht and Hb is increased because of spleen contraction. In addition, median age for each group should be added for AP interpretation.

The study has got a lot of limitation. For me the main thing is lack of pasture at two last months. Probably it had the strongest impact for hematological changes in examined horses.

Round 2

Reviewer 2 Report

The Authors corrected the manuscript.

However, I still recommend to add briefly some information about mentioned in the introdution bio-active componnents because in other case there is no need to mention them at all. After this correction the article can be considered to be published in Animals journal.